# RelA-Diffusion: Relativistic Adversarial Diffusion for Multi-Tracer PET Synthesis from Multi-Sequence MRI

## Abstract

Multi-tracer positron emission tomography (PET) provides critical insights into diverse neuropathological processes such as tau accumulation, neuroinflammation, and $\beta$-amyloid deposition in the brain, making it indispensable for comprehensive neurological assessment. However, routine acquisition of multi-tracer PET is limited by high costs, radiation exposure, and restricted tracer availability. Recent efforts have explored deep learning approaches for synthesizing PET images from structural MRI. While some methods rely solely on T1-weighted MRI, others incorporate additional sequences such as T2-FLAIR to improve pathological sensitivity. However, existing methods often struggle to capture fine-grained anatomical and pathological details, resulting in artifacts and unrealistic outputs. To address these limitations, we propose RelA-Diffusion, a Relativistic Adversarial Diffusion framework for multi-tracer PET synthesis from multi-sequence MRI. By leveraging both T1-weighted and T2-FLAIR scans as complementary inputs, RelA-Diffusion captures richer structural information to guide image generation. To improve synthesis fidelity, we introduce a gradient-penalized relativistic adversarial loss to the intermediate clean predictions of the diffusion model. This loss compares real and generated images in a relative manner, encouraging the synthesis of more realistic local structures. Both the relativistic formulation and the gradient penalty contribute to stabilizing the training, while adversarial feedback at each diffusion timestep enables consistent refinement throughout the generation process. Extensive experiments on two datasets demonstrate that RelA-Diffusion outperforms existing methods in both visual fidelity and quantitative metrics, highlighting its potential for accurate synthesis of multi-tracer PET.

## 1 Introduction

Accurate assessment and monitoring of neurodegenerative disorders rely heavily on the ability to capture diverse and subtle pathological processes within the brain. Brain positron emission tomography (PET) is a powerful imaging modality that enables the *in vivo* visualization of various molecular targets, including $\beta$-amyloid deposition with $^{11}$C-PIB (PIB), tau pathology with $^{18}$F-AV1451 (TAU) Johnson et al. (2016); Schwarz et al. (2016); Antoni et al. (2013); Xia et al. (2013), and neuroinflammatory activity with $^{18}$F-PBR111 (PBR) Colasanti et al. (2014). This multi-tracer PET imaging can provide complementary information, offering a more comprehensive understanding of disease mechanisms. However, the use of multi-tracer PET imaging in clinical and research settings is hindered by several practical constraints, including high scanning costs, radiation exposure, and limited access to specific tracers. These limitations have motivated the development of alternative approaches that can synthesize PET data without requiring tracer administration. Recent research has focused on developing deep learning techniques to synthesize PET images from commonly accessible structural MRI scans, offering a non-invasive and cost-effective option.

Earlier methods for MRI-based PET image synthesis primarily employed GAN-based models. Wei et al. (2020) propose a conditional flexible self-attention GAN model to predict PET parametric maps from multi-sequence MRI, training the GAN using a Sketcher–Refiner approach. Hu et al. (2021) introduce a bidirectional GAN that learns both PET generation and inversion back to latent representations, aiming to effectively capture semantic content within the latent space. Zhang

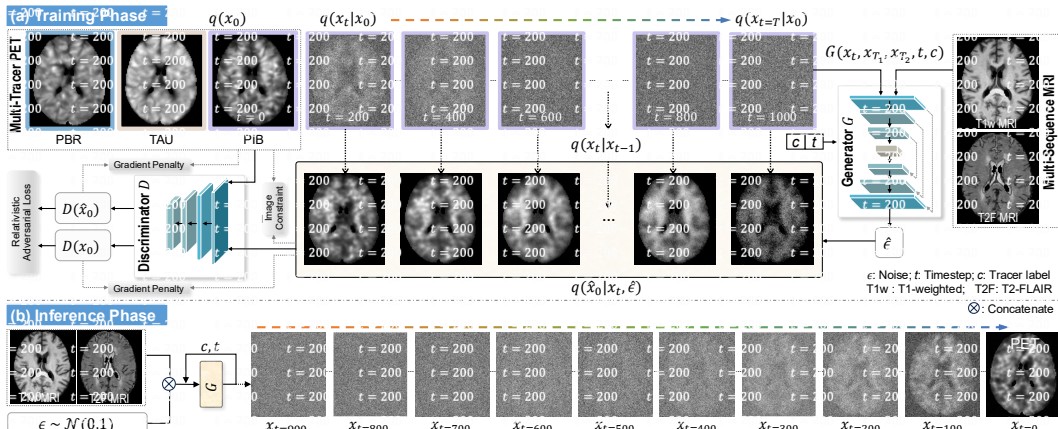

Figure 1: Illustration of the proposed relativistic adversarial diffusion (RelA-Diffusion) framework that synthesizes multi-tracer PET images from multi-sequence MRI input such as T1-weighted (T1w) and T2-FLAIR (T2F) MRIs.

et al. (2022) develop BPGAN, which utilizes a 3D multi-convolution U-Net generator with gradient profile and structural similarity index measure constraints to better preserve structural details in synthesized PET images. Several studies have applied CycleGAN-based frameworks for paired MRI-to-PET synthesis. For instance, Pan et al. (2021) introduce a method that translates between T1-weighted MRI and fluorodeoxyglucose-PET while integrating disease-specific feature disentanglement and diagnosis-guided supervision to enhance clinical relevance. Zhou et al. (2021) explore translation between multiple PET tracers, conditioning the synthesis process on anatomical context derived from structural MRI. Despite their effectiveness, GAN-based approaches frequently encounter optimization challenges, such as mode collapse, which can significantly limit model performance Goodfellow (2016).

More recent strategies have shifted toward diffusion models, which offer significant advantages in generating high-quality medical images. For example, a Joint Diffusion Attention Model (JDAM) (Xie et al., 2024) is designed to generate synthetic PET images from high-field and ultra-high-field MRI by learning the joint probability distribution between MRI input and the noisy PET output. Another study (Yu et al., 2024) presents a functional imaging constrained diffusion (FICD) method, incorporating a voxel-wise alignment loss to improve the quantitative fidelity of PET predictions by enforcing agreement with the ground truth. Additionally, a diffusion model MTGD (Zhong et al., 2025) is designed to fuse multi-sequence MRI to facilitate multi-tracer PET synthesis, showing improved performance in combining complementary MRI contrasts.

To leverage the strengths of both paradigms, several approaches have integrated GAN objectives into diffusion frameworks. The study by (Xiao et al., 2021) models each denoising step of a diffusion process using a multi-modal conditional GAN, allowing for significantly faster sampling while maintaining high sample quality and diversity, and improving mode coverage and training stability compared to traditional GANs. Another study (Wang et al., 2023) introduces Diffusion-GAN, a GAN framework that injects instance noise generated by a forward diffusion chain into the discriminator's input, thereby addressing GAN training instability and mode collapse by providing more stable and diverse learning signals. While these hybrid frameworks mitigate several limitations of standalone GANs and diffusion models, they often introduce complex training dynamics due to the competing nature of adversarial and denoising objectives. Consequently, such models may struggle to balance global structural coherence with fine-grained detail, leading to suboptimal synthesis fidelity in some cases.

To address these critical limitations, we propose RelA-Diffusion, a novel relativistic adversarial diffusion framework specifically developed to synthesize realistic multi-tracer PET images from multi-sequence MRI scans, including T1-weighted (T1w) and T2-FLAIR (T2F) MR images. As illustrated in Figure 1, RelA-Diffusion employs a conditional denoising diffusion probabilistic model that iteratively transforms Gaussian noise into realistic PET images, conditioned on multi-sequence MRI inputs. To improve training dynamics and enhance image fidelity, we introduce the gradient-

penalized relativistic adversarial loss during training. This loss directly compares real and synthesized PET images to encourage sharper anatomical boundaries and more realistic local structures. Both the gradient penalty and the relativistic formulation help stabilize adversarial learning and regularize model behavior, reducing artifacts and promoting detailed structural representation. Extensive evaluations on two multi-tracer PET datasets demonstrate that RelA-Diffusion consistently outperforms state-of-the-art (SOTA) methods in terms of visual realism, anatomical accuracy, and quantitative performance metrics, establishing its effectiveness and reliability for multi-tracer PET image synthesis. The main contributions of this work are summarized as follows:

- We introduce RelA-Diffusion, a diffusion-based framework tailored for multi-tracer PET synthesis from multi-sequence MRI, leveraging both T1-weighted and T2-FLAIR MRI inputs for enhanced pathological sensitivity.

- We propose to apply the gradient-penalized relativistic adversarial feedback to the intermediate clear image estimated during the training of the diffusion model, enabling consistent supervision and improved detail preservation, while stabilizing the training.

- Extensive experiments on two real-world datasets demonstrate that RelA-Diffusion consistently outperforms existing state-of-the-art methods in both visual fidelity and quantitative metrics.

## 2 PROPOSED METHOD

We propose **RelA-Diffusion**, a conditional diffusion framework for synthesizing high-fidelity multi-tracer PET images from multi-sequence MRI. The framework introduces a stabilized adversarial supervision mechanism to diffusion training by leveraging a gradient-penalized relativistic discriminator to provide feedback on intermediate clean predictions $\hat{x}_0$ at each training step. This feedback guides the learning of fine-grained tracer uptake patterns while ensuring stable optimization dynamics throughout the denoising process.

### 2.0.1 OVERALL FRAMEWORK.

RelA-Diffusion consists of two main components: a *conditional denoising diffusion model* as the generator $G$, and a *relativistic discriminator $D$* used only during training. As illustrated in Figure 1, the generator predicts the noise component $\hat{\epsilon} = G(x_t, x_{T_1}, x_{T_2}, t, c)$ at each denoising step $t$, where $x_t$ is the noisy image, $x_{T_1}$ and $x_{T_2}$ are the T1w and T2F MRI inputs, and $c$ denotes the tracer label. The clean image estimate $\hat{x}_0$ is then computed from the predicted noise and $x_t$ using the DDPM formulation (Ho et al., 2020). The discriminator $D(\hat{x}_0, x_0)$ compares the predicted $\hat{x}_0$ with the ground-truth PET image $x_0$ in a relativistic manner to provide adversarial feedback on the relative realism of the intermediate prediction. At inference time, only the trained generator $G$ is used to synthesize PET images through the learned reverse diffusion process, without involving the discriminator.

### 2.0.2 DIFFUSION PROCESS.

Our generative model follows a conditional denoising diffusion framework (Ho et al., 2020), which defines a forward noising process as a fixed Markov chain that progressively corrupts a clean PET image $x_0$ into a sequence of noisy variables $\{x_t\}_{t=1}^{T}$ over $T$ timesteps. This is achieved by adding Gaussian noise at each step according to a predefined noise schedule Ho et al. (2020). The reverse denoising process is then learned to invert this degradation, reconstructing the original data by estimating the noise added at each step.

### 2.0.3 FORWARD DIFFUSION PROCESS.

Given a real PET image $x_0 \sim q(x_0)$, the forward process adds Gaussian noise over $T$ timesteps, creating a series of noisy samples $\{x_1, \cdots, x_T\}$. This process is defined by a fixed Markov chain:

$$q(x_{1:T}|x_0) = \prod_{t=1}^{T} q(x_t|x_{t-1}), \qquad (1)$$

$$q(x_t|x_{t-1}) = \mathcal{N}(x_t; \sqrt{1-\beta_t}x_{t-1}, \beta_t\mathbf{I}), \qquad (2)$$

where $\mathcal{N}$ is Gaussian distribution with mean $\mu_t$ and variance $\sigma_t$, $\beta_t$ is a time-dependent hyperparameter that controls the noise level at each timestep, and $\mathbf{I}$ is the identity matrix indicating isotropic variance. The marginal distribution at timestep $t$ can be expressed as:

$$q(x_t|x_0) = \mathcal{N}(x_t; \sqrt{\bar{\alpha}_t}x_0, (1 - \bar{\alpha}_t)\mathbf{I}), \tag{3}$$

where $\alpha_t = 1 - \beta_t$ and $\bar{\alpha}_t = \prod_{s=1}^{t} \alpha_s$.

### 2.0.4 REVERSE DENOISING PROCESS.

Starting from a noisy image $x_t$, the reverse denoising process aims to progressively estimate and remove noise to recover the cleaner image $x_{t-1}$ in the previous timestep, and ultimately aims to reconstruct the original data $x_0$. The trainable component for this process is a generator $G$ that predicts the noise $\epsilon$ added to $x_0$ during the forward diffusion process to obtain $x_t$, denoted as $\hat{\epsilon}$. As illustrated in the top right of Figure 1, the generator is conditioned on multi-sequence MRI inputs (i.e., $x_{T_1}$ for T1w MRI and $x_{T_2}$ for T2F MRI) via input channel concatenation, and conditioned on the tracer label $c$ and timestep $t$ via cross-attention mechanisms, allowing the model to leverage anatomical information from MRI and the target tracer type to guide PET image synthesis.

In the *training phase*, a randomly selected timestep $t \in \{1, \cdots, T\}$ is sampled uniformly, and a noisy image $x_t$ is generated by adding Gaussian noise to the ground-truth PET image $x_0$ according to the forward diffusion process:

$$x_t = \sqrt{\bar{\alpha}_t}x_0 + \sqrt{1 - \bar{\alpha}_t}\,\epsilon, \quad \epsilon \sim \mathcal{N}(0, \mathbf{I}). \tag{4}$$

The generator $G$ is then trained to predict the added noise $\epsilon$ from the noisy image $x_t$. The training objective is to minimize the mean squared error (MSE) between the predicted noise $\hat{\epsilon}$ and the ground-truth noise $\epsilon$, defined as:

$$\mathcal{L}_N = \frac{1}{n} \sum_{i=1}^{n} (\epsilon - \hat{\epsilon}(x_t, t, x_{T_1}, x_{T_2}))^2, \tag{5}$$

where $n$ denotes the number of samples. Given this predicted noise, an estimate of the clean image $\hat{x}_0$ at timestep $t$ can be computed as:

$$\hat{x}_0(x_t, t) = \frac{x_t - \sqrt{1 - \bar{\alpha}_t}\,\hat{\epsilon}}{\sqrt{\bar{\alpha}_t}}. \tag{6}$$

In the *inference phase*, the generator iteratively applies its noise predictions to progressively denoise an initial pure Gaussian noise input, starting from $x_T$ and moving step-by-step towards $x_0$. At each step, the generator predicts the noise $\hat{\epsilon}$, which is then utilized to estimate a cleaner version:

$$x_{t-1} = \frac{1}{\sqrt{\alpha_t}} \left( x_t - \frac{1 - \alpha_t}{\sqrt{1 - \bar{\alpha}_t}}\hat{\epsilon} \right) + \sigma_t z, \tag{7}$$

where $\sigma_t$ is the standard deviation of added noise and $z \sim \mathcal{N}(0, 1)$. This iterative sampling procedure, guided by the estimated $\hat{x}_0$ and the conditional multi-sequence MRI and tracer label inputs, generating the multi-tracer PET image.

### 2.0.5 SUPERVISION ON INTERMEDIATE CLEAN PREDICTIONS.

To enhance output fidelity, we introduce the following two constraints on the intermediate clean image $\hat{x}_0$ by comparing them against the true $x_0$ in RelA-Diffusion.

(1) *Image Constraint*: The image-level constraint aims to minimize the $l_1$ loss between $\hat{x}_0$ and $x_0$, formulated as:

$$\mathcal{L}_I = \frac{1}{n} \sum_{i=1}^{n} |x_0 - \hat{x}_0|, \tag{8}$$

where $n$ denotes the number of samples. This design explicitly guides $\hat{x}_0$ during training to promote more reliable outputs across the entire reverse process. It is essential since denoising becomes increasingly difficult at larger timesteps, where the input $x_t$ is heavily corrupted. Supervising $\hat{x}_0$ at all timesteps implicitly places stronger regularization on the model's behavior in high-noise diffusion steps, enabling the network to recover meaningful details even from severely degraded inputs. This

effect has been theoretically discussed in prior works Hang et al. (2023); Yu et al. (2024) and provides a foundation for the subsequent adversarial feedback, ensuring that the discriminator receives informative and reliable intermediate predictions at all diffusion steps.

(2) **Gradient-Penalized Relativistic Adversarial Constraint**: To further enhance realism and detail of synthesized PET images, we integrate a *relativistic adversarial loss* Jolicoeur-Martineau (2019); Huang et al. (2024). Unlike traditional GANs where the discriminator judges whether an image is real or fake in an absolute manner, a relativistic discriminator assesses whether a generated image is more realistic than a real image, or vice versa. The discriminator $D$ is a classifier that receives both intermediate clean predictions $\hat{x}_0$ produced from noisy PET and the corresponding real PET image $x_0$. We perform pairwise comparison to estimate the relative authenticity between real and generated samples. Denote $\mathbb{P}$ and $\mathbb{Q}$ as the probability distributions of real and synthesized PET images, respectively. The relativistic adversarial (RA) loss is formulated as:

$$\mathcal{L}_R = \mathbb{E}_{(x_0, \hat{x}_0) \sim (\mathbb{P}, \mathbb{Q})} \left[ f(D(\hat{x}_0) - D(x_0)) \right]$$
$$+ \mathbb{E}_{(x_0, \hat{x}_0) \sim (\mathbb{P}, \mathbb{Q})} \left[ f(D(x_0) - D(\hat{x}_0)) \right], \tag{9}$$

where $D(x)$ denotes the discriminator's scalar output for image $x$, $D(\hat{x})$ denotes the output for the synthesized image $\hat{x}$, $f(y) = -\log(1 + e^{-y})$ is the activation function Nowozin et al. (2016), and the expectations are computed over the batch of real and synthesized images. The first term in Eq. equation 9 encourages the generator to increase the relative realism of its outputs compared to real PET images. The second term is designed to estimate how much more realistic a real image is compared to a generated one, rather than assigning a realism score to each image in isolation.

Compared to conventional GANs, which can suffer from gradient saturation and mode collapse due to the discriminator's binary real/fake decisions, the relativistic mechanism has been shown to provide a smoother learning signal that stabilizes adversarial training. It can also improve image quality by preventing the discriminator from becoming overly confident in real-vs-fake discrimination, to maintain meaningful gradients throughout training. Applied jointly with the diffusion loss, it promotes more consistent improvements in the perceptual quality of the diffusion model and guides the model toward producing more realistic outputs.

To facilitate model training, we also incorporate a zero-centered *gradient penalty* on both real and generated data Huang et al. (2024). Specifically, we penalize the squared norm of the discriminator's gradients with respect to its inputs. This penalty constrains the gradient norm of the discriminator $D$ on the real PET image $x_0$ and the generated image $\hat{x}_0$, which is formulated as:

$$\mathcal{L}_G = \mathbb{E}_{x_0 \sim \mathbb{P}} \left[ \|\nabla_{x_0} D(x_0)\|^2 \right]$$
$$+ \mathbb{E}_{\hat{x}_0 \sim \mathbb{Q}} \left[ \|\nabla_{\hat{x}_0} D(\hat{x}_0)\|^2 \right], \tag{10}$$

where $\nabla$ is the gradient operation. This gradient penalty (GP) is designed to promote smoother discriminator behavior on both real and synthesized PET images, improving convergence and ensuring stability throughout training Huang et al. (2024); Nagarajan & Kolter (2017).

### 2.0.6 OVERALL TRAINING OBJECTIVE.

The RelA-Diffusion framework is trained by minimizing a hybrid objective function that balances the diffusion model's noise prediction accuracy with the image constraint and the gradient-penalized adversarial guidance for realism. The hybrid loss $\mathcal{L}$ for RelA-Diffusion is finally formulated as:

$$\mathcal{L} = \mathcal{L}_N + \mathcal{L}_I + \mathcal{L}_R + \mathcal{L}_G. \tag{11}$$

### 2.0.7 IMPLEMENTATION.

The proposed RelA-Diffusion is implemented using the MONAI framework Cardoso et al. (2022) on PyTorch and trained on a computing cluster with multiple NVIDIA H100 GPUs (each with 80GB memory). We employ the Adam optimizer for both the generator and the discriminator. Empirically, the learning rate for the generator is set to $5 \times 10^{-5}$ and for the discriminator to $5 \times 10^{-6}$. Training is performed with a batch size of 3. The diffusion process is set to $T = 1000$ timesteps with a linear noise schedule for $\beta_t$ values, ranging from $\beta_1 = 0.0005$ to $\beta_T = 0.0195$. The model is trained for 100 epochs. The adversarial loss $\mathcal{L}_R + \mathcal{L}_G$ is weighted by 0.1 throughout all experiments.

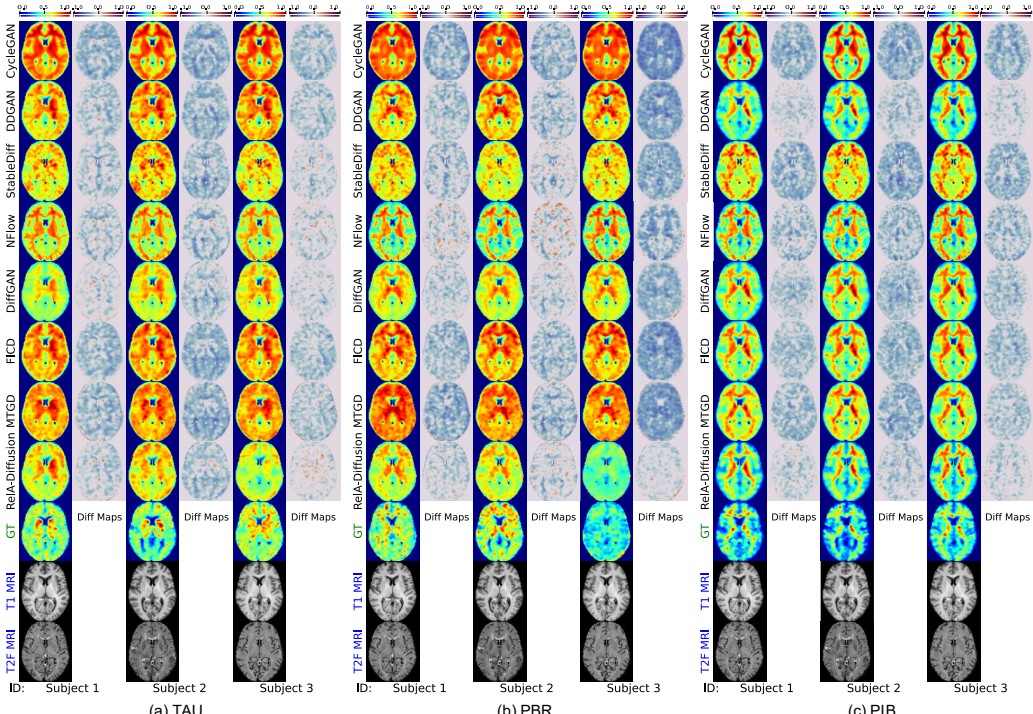

Figure 2: Visualization of test PET images for (a) TAU, (b) PBR, and (c) PIB, synthesized by 8 methods, along with corresponding difference (Diff) maps. Ground-truth (GT) and input T1w/T2F MRIs with subject IDs are shown at the bottom.

## 3 EXPERIMENTS

### 3.0.1 MATERIALS AND IMAGE PRE-PROCESSING.

Two datasets are utilized in this study: (1) NFL-LONG Walton et al. (2022) comprises 152 participants with a mean age of $60.15 \pm 6.25$ years. While all subjects have T1-weighted (T1w) and T2-FLAIR (T2F) MRI scans, the availability of PET tracers varies across individuals: 142 subjects have PBR-PET, 105 subjects have PIB-PET, and 124 subjects have TAU-PET using the AV1451 tracer. A subset of 80 subjects possesses all imaging modalities. From this complete cohort, 10 subjects are used for testing across all three tracer types, while the remaining data are used for training. Multi-tracer PET from the same session is rare in public datasets. (2) ADNI Jack Jr. et al. (2008), a public dataset, contains 502 cognitively normal subjects with available T1w MRI, T2F MRI, and TAU-PET scans (with the AV1451 tracer) in the baseline visit. We use all 502 subjects as an external evaluation cohort to assess the generalizability of our model. The mean age of these ADNI subjects is $69.83 \pm 8.80$. The subject IDs used are provided in the *Supplementary Materials* for reproducibility. T1w MRI scans are registered to the MNI space. The PET images are rigidly aligned to the corresponding T1w MRI in subject space and then nonlinearly warped to the MNI space using the MRI-derived deformation field. The T2-FLAIR images follow the same pipeline as PET, with an additional bias correction applied prior to rigid alignment. All volumes are resampled to an isotropic resolution of $1 \times 1 \times 1 \text{mm}^3$, and a standard MNI brain mask is applied for skull removal. To exclude irrelevant background, all images are center-cropped to $160 \times 180 \times 160$. During training, voxel intensities are scaled to the range $[-1, 1]$, while evaluation uses the normalized range $[0, 1]$ for both visual and quantitative analyses.

### 3.0.2 EXPERIMENTAL SETTINGS.

We quantitatively evaluate image quality using three metrics: peak signal-to-noise ratio (PSNR), structural similarity index (SSIM), and mean absolute error (MAE). Prior to evaluation, all synthesized PET images are intensity-normalized to $[0, 1]$ and padded to match their original spatial dimensions.

Table 1: Quantitative results of eight methods for TAU-, PBR- and PIB-PET generation on NFL-LONG (best results in bold). The column *sig* indicates statistically significant ($p < 0.05$).

| Method | Synthesized TAU-PET Image | | | | Synthesized PBR-PET Image | | | | Synthesized PIB-PET Image | | | |
|---|---|---|---|---|---|---|---|---|---|---|---|---|
| | PSNR↑ | SSIM↑ | MAE↓ | Sig. | PSNR↑ | SSIM↑ | MAE↓ | Sig. | PSNR↑ | SSIM↑ | MAE↓ | Sig. |
| CycleGAN (Zhou et al., 2021) | $27.506_{\pm1.598}$ | $0.869_{\pm0.013}$ | $\mathbf{0.017}_{\pm0.004}$ | * | $28.065_{\pm2.687}$ | $0.833_{\pm0.022}$ | $0.017_{\pm0.007}$ | * | $24.954_{\pm2.274}$ | $0.846_{\pm0.017}$ | $0.024_{\pm0.010}$ | |
| DDGAN (Xiao et al., 2021) | $25.448_{\pm2.373}$ | $0.892_{\pm0.015}$ | $0.025_{\pm0.009}$ | * | $25.201_{\pm3.044}$ | $0.884_{\pm0.017}$ | $0.027_{\pm0.011}$ | * | $\mathbf{26.382}_{\pm2.561}$ | $\mathbf{0.870}_{\pm0.018}$ | $\mathbf{0.021}_{\pm0.010}$ | * |
| StableDiff (Rombach et al., 2022) | $26.483_{\pm4.426}$ | $0.881_{\pm0.031}$ | $0.023_{\pm0.015}$ | * | $\mathbf{28.845}_{\pm2.681}$ | $0.892_{\pm0.019}$ | $\mathbf{0.015}_{\pm0.006}$ | * | $25.245_{\pm1.735}$ | $0.852_{\pm0.013}$ | $0.023_{\pm0.007}$ | * |
| NFlow (Beizaee et al., 2023) | $27.615_{\pm3.490}$ | $0.881_{\pm0.020}$ | $0.019_{\pm0.010}$ | * | $28.844_{\pm2.030}$ | $0.887_{\pm0.013}$ | $\mathbf{0.015}_{\pm0.004}$ | * | $24.522_{\pm1.304}$ | $0.805_{\pm0.017}$ | $0.024_{\pm0.004}$ | * |
| DiffGAN (Wang et al., 2023) | $24.980_{\pm2.330}$ | $0.884_{\pm0.013}$ | $0.026_{\pm0.009}$ | * | $23.112_{\pm2.938}$ | $0.870_{\pm0.021}$ | $0.034_{\pm0.012}$ | * | $23.491_{\pm2.640}$ | $0.855_{\pm0.021}$ | $0.031_{\pm0.013}$ | * |
| FICD (Yu et al., 2024) | $24.236_{\pm2.877}$ | $0.883_{\pm0.015}$ | $0.030_{\pm0.012}$ | * | $25.594_{\pm2.945}$ | $0.885_{\pm0.017}$ | $0.025_{\pm0.010}$ | * | $24.229_{\pm2.860}$ | $0.860_{\pm0.021}$ | $0.029_{\pm0.012}$ | * |
| MTGD (Zhong et al., 2025) | $24.998_{\pm1.937}$ | $0.872_{\pm0.012}$ | $0.026_{\pm0.008}$ | * | $23.847_{\pm2.379}$ | $0.859_{\pm0.012}$ | $0.031_{\pm0.010}$ | * | $25.677_{\pm2.326}$ | $0.846_{\pm0.013}$ | $0.023_{\pm0.010}$ | * |
| RelA-Diffusion (Ours) | $\mathbf{28.314}_{\pm3.392}$ | $\mathbf{0.898}_{\pm0.017}$ | $\mathbf{0.017}_{\pm0.009}$ | – | $\mathbf{29.324}_{\pm2.437}$ | $\mathbf{0.898}_{\pm0.017}$ | $\mathbf{0.015}_{\pm0.006}$ | – | $26.270_{\pm1.687}$ | $0.861_{\pm0.012}$ | $\mathbf{0.020}_{\pm0.006}$ | – |

### 3.0.3 COMPETING METHODS.

We compare our method against seven SOTA methods for medical image synthesis. (1) **Cycle-GAN** Zhou et al. (2021): A conditional GAN that concatenates tracer labels with the input and employs cycle-consistency loss alongside adversarial and $l_1$ losses. (2) Denoising Diffusion GAN (**DDGAN**) Xiao et al. (2021): This method models each denoising step of a diffusion process using a multi-modal conditional GAN, allowing for faster sampling while maintaining high sample quality and diversity by improving mode coverage and training stability. (3) Stable Diffusion (**StableDiff**) Rombach et al. (2022): A latent diffusion model that encodes 3D images into a compact latent space before applying diffusion-based modality translation, and then uses a decoder to recover the image from the latent space. Its encoder and decoder components are pretrained on all training PET images in this study and kept frozen during the diffusion model's training for the modality translation task. (4) Normalizing Flow (**NFlow**) Beizaee et al. (2023): A normalizing flow-based model trained end-to-end, leveraging normalizing flows to guide PET synthesis. (5) Diffusion GAN (**DiffGAN**) Wang et al. (2023): This GAN framework injects instance noise, generated by a forward diffusion chain, into the discriminator's input. This approach aims to address GAN training instability and mode collapse by providing more stable and diverse learning signals. (6) **FICD** Yu et al. (2024): A denoising diffusion probabilistic model that learns a reverse denoising process within a Markov chain and constrained by the clear intermediate prediction. (7) **MTGD** Zhong et al. (2025): A diffusion-based model for multi-tracer PET synthesis from multi-sequence MRI (T1w, T2w, T2F), using a Multi-Sequence Attention Encoder and cross-attention fusion. The fused representation, combined with timestep and tracer label, conditions the U-Net via its bottleneck. To ensure a fair comparison, all methods are implemented on 3D images, with architectures and hyperparameters kept as consistent as possible. All diffusion models operated at the image level (i.e., FICD, DDGAN, DiffGAN, and MTGD) implement the image constraint $\mathcal{L}_I$ supervision on intermediate clean predictions as standard DDPM often fails to capture fine structural details in 3D settings. All diffusion models (i.e., DDGAN, StableDiff, DiffGAN, FICD, and MTGD) receive input by concatenating T1w and T2F MRI and noisy PET along the channel dimension, and synthesize tracer-specific PET based on the input label. The tracer label is embedded together with the diffusion timestep and injected into the network. CycleGAN and NFlow input concatenated T1w and T2F MRI and generate three-channel outputs, each representing a tracer-specific PET image.

### 3.0.4 QUALITATIVE RESULTS.

Figure 2 displays axial PET slices synthesized by RelA-Diffusion and seven competing methods on the NFL-LONG dataset, alongside the corresponding ground-truth PET images and input T1w and T2F MRIs shown at the bottom. For each method, we also visualize the difference (Diff) maps between its synthesized PET and the ground truth to highlight voxel-level deviations. As shown in Figure 2, our RelA-Diffusion consistently produces PET images with higher visual quality and closer resemblance to the ground truth compared to the competing approaches. This improvement is particularly evident in the PBR and PIB tracers, where our method effectively captures subject-specific patterns of neuroinflammation and amyloid deposition in the brain. In contrast, the competing methods tend to generate similar outputs across different subjects, failing to reflect individual biological variation. Additionally, hybrid methods that combine diffusion with adversarial learning (e.g., DiffGAN and Rela-Diffusion) outperform those relying solely on GANs (e.g., CycleGAN) or diffusion models (e.g., FICD, MTGD, StableDiff). This highlights the benefit of integrating adversarial supervision into the diffusion framework to improve PET image synthesis. Among the hybrid methods, our RelA-Diffusion yields the best visual outputs.

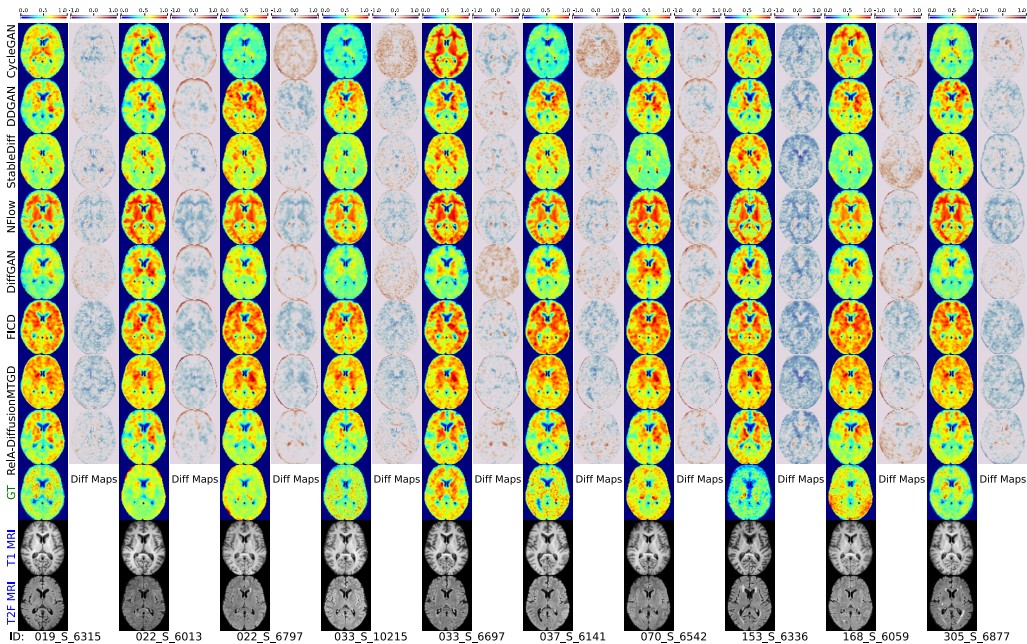

Figure 3: Visualization of test ADNI TAU-PET images synthesized by 8 methods, and difference (Diff) maps. The ground-truth (GT) PET images along with the input T1w and T2F MRIs are displayed at the bottom with the corresponding subject IDs.

Table 2: Quantitative results of eight methods for synthesized ADNI TAU-PET images (best results in bold).

| Method | PSNR↑ | SSIM↑ | MAE↓ |
|---|---|---|---|
| CycleGAN (Zhou et al., 2021) | 23.766±3.170 | 0.843±0.034 | 0.030±0.015 |
| DDGAN (Xiao et al., 2021) | 20.563±3.126 | 0.839±0.028 | 0.045±0.018 |
| StableDiff (Rombach et al., 2022) | 22.179±3.279 | 0.838±0.035 | 0.038±0.018 |
| NFlow (Beizaee et al., 2023) | **25.449**±2.635 | 0.851±0.033 | **0.024**±0.010 |
| DiffGAN (Wang et al., 2023) | 24.797±2.887 | 0.862±0.029 | 0.026±0.011 |
| FICD (Yu et al., 2024) | 18.624±2.934 | 0.831±0.028 | 0.058±0.021 |
| MTGD (Zhong et al., 2025) | 17.632±2.374 | 0.818±0.023 | 0.064±0.019 |
| RelA-Diffusion (Ours) | 23.781±3.993 | **0.864**±0.034 | 0.032±0.015 |

### 3.0.5 QUANTITATIVE RESULTS.

Table 1 reports the quantitative performance of all methods on the NFL-LONG cohort. Across all three PET tracers, RelA-Diffusion consistently achieves the best or highly competitive results. For TAU-PET synthesis, it achieves the highest PSNR (28.314) and SSIM (0.898), and ties with CycleGAN for the lowest MAE (0.017), indicating accurate voxel-wise reconstruction and strong anatomical preservation. For PBR-PET, RelA-Diffusion again achieves the best performance with a PSNR of 29.324, SSIM of 0.898, and MAE of 0.015, outperforming StableDiff and NFlow. While the latter show competitive MAE values, their slightly lower PSNR and SSIM indicate that RelA-Diffusion captures tracer patterns more faithfully. For PIB-PET synthesis, while DDGAN obtains the highest SSIM (0.870), RelA-Diffusion still delivers the best MAE (0.020) and a competitive PSNR (26.270), outperforming other diffusion-based methods (e.g., FICD and MTGD).

## 4 DISCUSSION

### 4.0.1 GENERALIZATION TO EXTERNAL DATA.

To assess the generalization capability of our method, we perform external evaluation on ADNI Jack Jr. et al. (2008), which includes paired T1w and T2F MRIs along with TAU-PET scans acquired using the AV1451 tracer. Without any fine-tuning, the model trained on NFL-LONG is

Table 3: Quantitative results achieved by RelA-Diffusion, as well as their degraded variants for multi-tracer PET generation on NFL-LONG, with best results shown in bold.

| Method | Synthesized TAU-PET Image | | | Synthesized PBR-PET Image | | | Synthesized PIB-PET Image | | |
|---|---|---|---|---|---|---|---|---|---|
| | PSNR↑ | SSIM↑ | MAE↓ | PSNR↑ | SSIM↑ | MAE↓ | PSNR↑ | SSIM↑ | MAE↓ |
| w/oRA | $28.156_{\pm4.304}$ | $\mathbf{0.908}_{\pm0.023}$ | $0.019_{\pm0.012}$ | $29.311_{\pm2.547}$ | $\mathbf{0.912}_{\pm0.018}$ | $0.015_{\pm0.006}$ | $25.408_{\pm2.182}$ | $\mathbf{0.877}_{\pm0.010}$ | $0.022_{\pm0.006}$ |
| w/oGP | $18.401_{\pm2.780}$ | $0.780_{\pm0.027}$ | $0.061_{\pm0.021}$ | $20.947_{\pm2.732}$ | $0.786_{\pm0.029}$ | $0.045_{\pm0.013}$ | $16.855_{\pm0.953}$ | $0.764_{\pm0.008}$ | $0.066_{\pm0.007}$ |
| w/oRAGP | $24.956_{\pm4.644}$ | $0.877_{\pm0.038}$ | $0.029_{\pm0.016}$ | $29.261_{\pm4.173}$ | $0.900_{\pm0.029}$ | $0.016_{\pm0.010}$ | $23.721_{\pm3.166}$ | $0.850_{\pm0.020}$ | $0.029_{\pm0.010}$ |
| w/oT2F | $28.109_{\pm4.457}$ | $0.898_{\pm0.029}$ | $0.019_{\pm0.013}$ | $29.490_{\pm2.380}$ | $0.905_{\pm0.017}$ | $\mathbf{0.014}_{\pm0.005}$ | $25.366_{\pm1.814}$ | $0.863_{\pm0.012}$ | $0.022_{\pm0.005}$ |
| w/oT1w | $26.022_{\pm4.721}$ | $0.887_{\pm0.035}$ | $0.026_{\pm0.016}$ | $\mathbf{29.894}_{\pm2.839}$ | $0.907_{\pm0.020}$ | $\mathbf{0.014}_{\pm0.006}$ | $23.695_{\pm2.748}$ | $0.851_{\pm0.014}$ | $0.029_{\pm0.008}$ |
| RelA-Diffusion | $\mathbf{28.314}_{\pm3.392}$ | $0.898_{\pm0.017}$ | $\mathbf{0.017}_{\pm0.009}$ | $29.324_{\pm2.437}$ | $0.898_{\pm0.017}$ | $0.015_{\pm0.006}$ | $\mathbf{26.270}_{\pm1.687}$ | $0.861_{\pm0.012}$ | $\mathbf{0.020}_{\pm0.006}$ |

Table 4: Quantitative results and inference time of RelA-Diffusion and RelA-Diffusion-Latent for TAU-, PBR-, and PIB-PET generation on NFL-LONG.

| Method | Synthesized TAU-PET Image | | | Synthesized PBR-PET Image | | | Synthesized PIB-PET Image | | | Time (s/vol)↓ |
|---|---|---|---|---|---|---|---|---|---|---|
| | PSNR↑ | SSIM↑ | MAE↓ | PSNR↑ | SSIM↑ | MAE↓ | PSNR↑ | SSIM↑ | MAE↓ | |
| RelA-Diffusion (Ours) | $28.314_{\pm3.392}$ | $0.898_{\pm0.017}$ | $0.017_{\pm0.009}$ | $29.324_{\pm2.437}$ | $0.898_{\pm0.017}$ | $0.015_{\pm0.006}$ | $26.270_{\pm1.687}$ | $0.861_{\pm0.012}$ | $0.020_{\pm0.006}$ | 126.778 |
| RelA-Diffusion-Latent (Ours) | $27.921_{\pm3.902}$ | $0.900_{\pm0.022}$ | $0.019_{\pm0.011}$ | $28.823_{\pm1.395}$ | $0.901_{\pm0.012}$ | $0.015_{\pm0.004}$ | $24.523_{\pm2.076}$ | $0.865_{\pm0.010}$ | $0.024_{\pm0.005}$ | 0.778 |

directly applied to 30 ADNI subjects for TAU-PET synthesis. Table 2 and Figure 3 present the qualitative examples and quantitative metrics for multi-sequence MRI to TAU-PET synthesis on ADNI.

As shown in Table 2, our method consistently achieves the best performance across all metrics when compared with the seven SOTA methods, demonstrating superior generalization capability. Diffusion-only methods (e.g., FICD and MTGD) cannot yield good results compared to hybrid approaches such as DiffGAN, suggesting limited generalizability to unseen data. The qualitative results in Figure 3 further support this observation, where RelA-Diffusion produces images that closely resemble the ground truth, and the corresponding difference (Diff) maps are visibly lighter. These findings collectively highlight the strong generalizability of our method when applied to external datasets acquired from different scanners and scanning protocols, suggesting its greater potential for clinical deployment across diverse imaging environments and patient populations.

### 4.0.2 LATENT RELA-DIFFUSION.

We further develop a latent-space variant of our framework, termed RelA-Diffusion-Latent, to improve computational efficiency. Instead of directly operating in the voxel space, both the conditional MRI inputs and the target PET images are first encoded into a latent space ($10\times12\times10$) using a pretrained frozen autoencoder. The diffusion model operates entirely in this latent space and predicts the noise added to the PET latent. The diffusion processes and the relativistic adversarial supervision remain the same as RelA-Diffusion. We apply the image constraint $L_I$ directly on the predicted clean latent $\hat{z}_0$ to enforce accurate reconstruction in the latent domain. In contrast, the gradient-penalized relativistic adversarial constraint is applied on the *decoded* PET image $\hat{x}_0$, obtained by passing $\hat{z}_0$ through the decoder, allowing the discriminator to assess perceptual realism in the image space. This hybrid latent–image supervision strategy preserves the computational advantages of latent diffusion while retaining strong adversarial guidance for high-fidelity PET synthesis. Table 4 compares RelA-Diffusion and RelA-Diffusion-Latent, showing comparable synthesis performance while the latent version significantly reduces inference time. Notably, RelA-Diffusion-Latent achieves an inference speed of 0.778 s/vol, which is even faster than the GAN baseline (CycleGAN: 1.47 s/vol), while diffusion-only methods such as FICD remain much slower (134.8 s/vol). This demonstrates that the latent variant offers an excellent efficiency–fidelity trade-off for practical PET synthesis.

### 4.0.3 ABLATION STUDY.

To evaluate the contributions of key components in our RelA-Diffusion framework, we conduct an ablation study on the NFL-LONG dataset by comparing it to four ablated variants: **(1) w/oRA**, which replaces the relativistic adversarial loss with a standard non-relativistic GAN loss; **(2) w/oGP**, which removes gradient penalties; **(3) w/oRAGP**, which replaces the relativistic adversarial loss with a standard non-relativistic GAN loss and removes gradient penalties; **(4) w/oT2F**, which excludes T2-

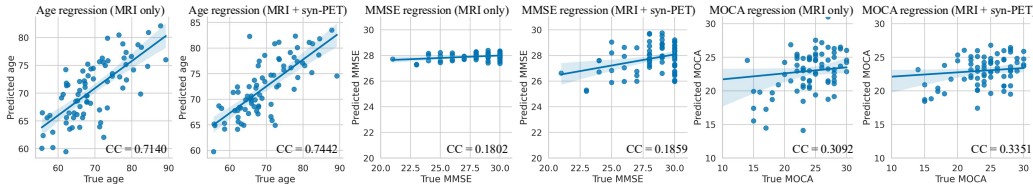

Figure 4: Downstream regression performance for age, MMSE, and MoCA prediction using synthesized PET (syn-PET) generated from ADNI TAU images together with MRI, in comparison with MRI-only inputs.

FLAIR input and uses only T1w MRI as condition; and **(5) w/oT1w**, which uses only T2-FLAIR MRI and discards T1w input. All variants retain the same network architecture as the full model to ensure a fair comparison. The quantitative results are reported in Table 3.

Ablation results in Table 3 demonstrate that removing gradient penalty (GP) leads to the largest performance drop across all tracers, confirming that GP is essential for stabilizing adversarial training and preventing degradation in diffusion–GAN synthesis. Replacing the relativistic adversarial loss (RA) with a standard GAN loss also reduces fidelity, indicating that RA further enhances structural consistency. When both replacing RA with a standard GAN loss and removing GP (w/oRAGP), there is a consistent drop in PSNR and increased MAE across all tracer types, demonstrating that GP and RA are complementary, and their combination is critical for achieving high-quality PET synthesis. Excluding T1w input (w/oT1w) degrades performance, particularly for TAU- and PBR-PET, indicating that the complementary anatomical context captured by T2-FLAIR and T1w MRIs is critical for accurate tracer distribution modeling. Similarly, removing T2-FLAIR (w/oT2F) also causes a performance drop, but the effect is slightly less pronounced than w/oT1w, suggesting that while both modalities contribute valuable information, T1w may offer more specific pathological contrast for PET synthesis. Overall, the full model leveraging both T1w and T2F MRIs with relativistic adversarial supervision achieves the best performance.

### 4.0.4 DOWNSTREAM TASKS.

To evaluate the clinical utility of the synthesized PET images, we further conduct downstream regression tasks for predicting age, Mini-Mental State Examination (MMSE) and Montreal Cognitive Assessment (MoCA) scores. We compare two input settings: (1) using MRI only, and (2) using MRI combined with synthesized PET (syn-PET) generated from ADNI TAU images. All regression models share the same ResNet-based architecture and training protocol to ensure a fair comparison. The results in Figure 4 demonstrate that incorporating syn-PET consistently improves prediction performance across all three tasks, indicating that the synthesized PET provides complementary functional information beyond structural MRI and is beneficial for clinically relevant cognitive outcome prediction.

## 5 CONCLUSION

This paper presents RelA-Diffusion, a novel relativistic adversarial diffusion framework for synthesizing multi-tracer PET images from multi-sequence MRI. By leveraging T1w and T2-FLAIR inputs, our method captures complementary structural and pathological cues to guide the generation process. The proposed gradient-penalized relativistic adversarial loss enhances anatomical realism and stabilizes training. Extensive experiments demonstrate that RelA-Diffusion achieves superior performance in both visual fidelity and quantitative accuracy compared to existing methods. This highlights its potential to reduce reliance on expensive, high-risk multi-tracer PET imaging in clinical and research settings. In future work, we plan to extend RelA-Diffusion to handle other imaging modalities, such as CT or contrast-enhanced MRI, to support broader clinical applications. We also aim to explore domain adaptation strategies to improve generalizability across diverse scanners.

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

# A APPENDIX

## A.1 RELA-DIFFUSION ALGORITHM IMPLEMENTATION

### A.1.1 TRAINING PROCEDURE (ALGORITHM 1)

The training procedure of RelA-Diffusion is outlined in Algorithm 1. The model is trained on paired multi-sequence MRI $(x_{T_1}, x_{T_2})$ and multi-tracer PET $x_0$ data. At each iteration, a timestep $t$ is sampled and Gaussian noise is added to $x_0$ to obtain a noisy input $x_t$. The generator $G$ takes $x_t$, MRI inputs, $t$, and tracer label $c$ to predict the noise $\hat{\epsilon}$, from which the clean estimate $\hat{x}_0$ is reconstructed. The relativistic adversarial loss uses the activation $f(x) = -\log(1 + e^{-x})$.

---

**Algorithm 1** Training Procedure for RelA-Diffusion

---

**Input**: Paired data $\{x_{T_1}, x_{T_2}, x_0, c\}_{i=1}^N$
**Parameter**: Diffusion steps $T$, learning rate $\eta$
**Output**: Trained generator $G$

1: Initialize generator $G$, discriminator $D$, and optimizers
2: **for** epoch = 1 to $N_{\text{epochs}}$ **do**
3:   **for** each batch $(x_{T_1}, x_{T_2}, x_0, c)$ **do**
4:     Sample timestep $t \sim \{1, \ldots, T\}$ uniformly
5:     Sample noise $\epsilon \sim \mathcal{N}(0, \mathbf{I})$
6:     Generate noisy PET: $x_t = \sqrt{\bar{\alpha}_t}x_0 + \sqrt{1 - \bar{\alpha}_t}\,\epsilon$
7:     Concatenate $x_t$, $x_{T_1}$, and $x_{T_2}$ along the channel dimension as model input
8:     Embed timestep $t$ and tracer label $c$; inject into $G$
9:     Predict noise: $\hat{\epsilon} = G(x_t, x_{T_1}, x_{T_2}, t, c)$
10:    Compute $\hat{x}_0 = \frac{x_t - \sqrt{1 - \bar{\alpha}_t}\hat{\epsilon}}{\sqrt{\bar{\alpha}_t}}$
11:    **Noise loss:** $\mathcal{L}_N = \frac{1}{n}\sum_{i=1}^n (\epsilon - \hat{\epsilon})^2$
12:    **Image loss:** $\mathcal{L}_I = \frac{1}{n}\sum_{i=1}^n |x_0 - \hat{x}_0|$
13:    **Relativistic adversarial loss:** $\mathcal{L}_R = f(D(\hat{x}_0) - D(x_0)) + f(D(x_0) - D(\hat{x}_0))$
14:    **Gradient penalty:** $\mathcal{L}_G = \|\nabla_{x_0} D(x_0)\|^2 + \|\nabla_{\hat{x}_0} D(\hat{x}_0)\|^2$
15:    **Total loss:** $\mathcal{L} = \mathcal{L}_N + \mathcal{L}_I + \mathcal{L}_R + \mathcal{L}_G$
16:    Update generator $G$ using $\nabla\mathcal{L}$
17:    Update discriminator $D$ using $\nabla\mathcal{L}$
18:   **end for**
19: **end for**
20: **return** trained generator $G$

---

### A.1.2 INFERENCE PROCEDURE (ALGORITHM 2)

Algorithm 2 describes the inference process for RelA-Diffusion. Given a trained generator $G$, multi-sequence MRI inputs $x_{T_1}$ and $x_{T_2}$, and a target tracer label $c$, the model synthesizes a PET image via an iterative denoising process. Starting from a pure Gaussian noise image $x_T \sim \mathcal{N}(0, \mathbf{I})$, the model runs the reverse diffusion process for $T$ steps. At each step $t$, the generator predicts the noise $\hat{\epsilon}$, and uses it to estimate a cleaner image $x_{t-1}$. This iterative refinement continues until $x_0$ is produced.

---

**Algorithm 2** Inference with Trained RelA-Diffusion Generator

---

**Input**: Multi-sequence MRI inputs $(x_{T_1}, x_{T_2})$, tracer label $c$, trained generator $G$
**Parameter**: Diffusion steps $T$, noise schedule $\{\alpha_t, \bar{\alpha}_t, \sigma_t\}_{t=1}^T$
**Output**: Synthesized PET image $\hat{x}_0$

 1: Initialize $x_T \sim \mathcal{N}(0, \mathbf{I})$            {Start from pure Gaussian noise}
 2: **for** $t = T$ to 1 **do**
 3:     Predict noise: $\hat{\epsilon} = G(x_t, x_{T_1}, x_{T_2}, t, c)$
 4:     Sample noise $z \sim \mathcal{N}(0, \mathbf{I})$ if $t > 1$, else $z = 0$
 5:     Compute denoised image:

$$x_{t-1} = \frac{1}{\sqrt{\alpha_t}} \left( x_t - \frac{1 - \alpha_t}{\sqrt{1 - \bar{\alpha}_t}} \hat{\epsilon} \right) + \sigma_t z$$

 6: **end for**
 7: **return** final image $\hat{x}_0$

---

EXTERNAL TEST SUBJECT IDS

The following subjects from ADNI were used to test the generalization ability of the proposed method in our experiments:

| | | | | | |
|---|---|---|---|---|---|
| 153_S_6450 | 033_S_10144 | 141_S_6872 | 301_S_6592 | 033_S_6697 | 006_S_6770 |
| 022_S_6797 | 941_S_6803 | 067_S_7061 | 037_S_6216 | 033_S_10215 | 305_S_6877 |
| 153_S_6336 | 141_S_6240 | 033_S_6889 | 941_S_6581 | 037_S_6141 | 033_S_10021 |
| 067_S_6117 | 301_S_6615 | 305_S_6498 | 019_S_6315 | 070_S_6542 | 016_S_6773 |
| 033_S_7114 | 168_S_6059 | 003_S_6606 | 067_S_6957 | 168_S_6085 | 022_S_6013 |

## A.2 NETWORK ARCHITECTURE.

(1) The generator is implemented using a U-Net architecture, consisting of 3 downsampling and 3 upsampling blocks. Each downsampling block contains one convolutional layer for spatial reduction and two residual sub-blocks incorporating skip connections. Each residual sub-block consists of two convolutional layers, each followed by group normalization and timestep embedding, along with a shortcut connection. Positional embeddings for the diffusion timestep $t$ plus tracer label $c$ project encoded features to match the number of feature channels, enabling effective feature integration. The multi-sequence MRI inputs are concatenated along the channel dimension and then fed into the U-Net along with the input $x_t$. (2) The discriminator is based on a PatchGAN discriminator architecture Wang et al. (2018), designed to operate on local image patches rather than the entire image. It consists of 3 convolutional layers with decreasing spatial resolution and increasing channel depth. Each convolutional layer is followed by LeakyReLU activation (with a negative slope of 0.2) and batch normalization, except for the final output layer. The output of the discriminator is a single scalar representing the relative realism of the input.

