# OpenReview forum: "RelA-Diffusion: Relativistic Adversarial Diffusion for Multi-Tracer PET Synthesis from Multi-Sequence MRI"
_ICLR.cc/2026/Conference — Submitted to ICLR 2026_

### Official Review · Reviewer_T9B3 · 2025-10-23

**Soundness:** 2
**Presentation:** 1
**Contribution:** 2
**Rating:** 2
**Confidence:** 4

**Summary:**

In this paper, the authors propose RelA-Diffusion: a 3D PET synthesis algorithm for multi-sequence MRI within the DDPM framework. The core approach is to simultaneously apply an L1 image constraint and a relative adversarial loss to the intermediate clean predictions $\hat{x_0}$ at each randomly sampled time step during training, with a zero-centered gradient penalty added to stabilize training. During inference, only the diffusion generator is used (the discriminator is not involved). On two datasets, the algorithm is compared with seven representative methods and achieves superior PSNR, SSIM, and MAE performance in most cases. Ablation results for de-modulation and de-modalization are also provided.

**Strengths:**

1. great motivation: PET is expensive and has radiation protection issues, so the generation of MRI-synthesized PET is a practical alternative.

2. clear method coupling: Relative discrimination and gradient penalty are applied to $\hat{x_0}$, forming a hybrid loss function L with the diffuse noise prediction and image L1.

**Weaknesses:**

1. The novelty is incremental, the main contribution seems like applying Diffusion-GAN to the generation from MRI to PET. It is suggested to focus on more concrete problem in the generation of PET.

2. It is suggested to add more downstream tasks to verify the performance of generation, like segmentation, classification.

3. In Table 1, CycleGAN outperforms DiffGAN, FICD, MTGD in most metrics across three tasks. Besides, the CycleGAN do not use the diffusion model. I wonder why the CycleGAN achieve so superior results? It is suggested to add more discussion about it, more importantly, the proposed method is also leveraging the diffusion model.

4. In formula 11, does the combination of losses need weights?

5. Many typos. In Line 40, there are double "with". In line 605, the paper format is confusing with "[!htp]".

**Questions:**

See Weakness

---

### Official Review · Reviewer_22Ei · 2025-10-28

**Soundness:** 3
**Presentation:** 3
**Contribution:** 3
**Rating:** 6
**Confidence:** 4

**Summary:**

This paper proposes RelA-Diffusion, a diffusion-based framework for synthesizing multi-tracer PET images (e.g., TAU, PIB, PBR) from multi-sequence MRI (T1w and T2-FLAIR). The key novelty lies in: 1) A gradient-penalized relativistic adversarial loss applied to intermediate clean predictions within the diffusion process. 2) Leveraging multi-sequence MRI for richer anatomical context. 3) Using relativistic comparison instead of binary real/fake judgments to stabilize training. Experiments on NFL-LONG and ADNI datasets demonstrate quantitative superiority over GAN, diffusion, and hybrid baselines.

**Strengths:**

The integration of relativistic adversarial loss into the diffusion framework is a novel way to stabilize adversarial training while improving structural fidelity.

The method section provides a clear mathematical formulation for forward/reverse diffusion, adversarial losses, and the hybrid objective. It is easy to follow.

Ablation experiments (w/o RA, w/o T1w, w/o T2F) convincingly support each component’s contribution. The method achieves consistent improvements in PSNR, SSIM, and MAE across three tracers and generalizes well to ADNI without fine-tuning. Visualizations (difference maps) clearly show sharper anatomical details and fewer artifacts.

**Weaknesses:**

The training involves 1000 diffusion steps and adversarial updates. No runtime, training time, or sampling speed comparison is provided against standard diffusion or GAN models.

Only 10 subjects for testing in NFL-LONG and 30 ADNI subjects for external validation are relatively small sample sizes, which might limit claims of robustness and clinical readiness.

**Questions:**

n/a

---

### Official Review · Reviewer_B64g · 2025-11-01

**Soundness:** 3
**Presentation:** 4
**Contribution:** 3
**Rating:** 4
**Confidence:** 4

**Summary:**

This paper presents a method for synthesizing PET images from structural MRI (T1w and T2-FLAIR). They propose technical innovations including a relativistic adversarial loss and gradient penalty to have sharper output images and capture fine-grained anatomical and pathological details. The authors evaluate on two datasets, training on and one and testing generalization on the other. They compare against several baselines and outperform in both qualitative and quantitative metrics.

**Strengths:**

The paper is clearly written, well-motivated, and extensively evaluated. They tackle an important problem, synthesizing PET images, which are both costly to obtain and patients often do not want to expose themselves to the radiation involved. As PET images are used for monitoring neurodegenerative disroders, synthetic images may improve the accuracy of predictions of disorder.

The authors propose a model that combines the strengths of both diffusion and GAN models to synthesize realistic images. They propose an extension to the standard diffusion framework by training with a relativistic adversarial loss and a gradient penalty. Ablations demonstrate that these extensions improve image synthesis quality.

They offer extensive experimental evaluations against 7 recent methods for this task. Qualitative evaluations also visually demonstrate the performance of their model.

**Weaknesses:**

### Clinical Utility
While I believe this paper is clearly written and technically sound, I have a hard time justifying the need for such a method. The authors claim that synthesizing PET images can be useful for clinical analysis of neurological disorder. However, I have a hard time believing that clinicians would actually use these types of images to make clinical decisions. While the authors evaluated well that the synthesized images were high quality, I am unconvinced that such a model would actually be used. A more interesting and useful experiment would be to demonstrate several tasks where training with the synthetic data improves performance on clinical classification, regression, or segmentation tasks over using just the T1w and T2F images.

Further, I have several small concerns that can be addressed.

### Evaluation and Generalization
The authors claim a key result is the ability of their model to generalize, by showing performance on 30 subjects from the ADNI dataset. There are a few issues here. First, 30 subjects is quite small, and I wonder why the authors did not just use the full 502 images in the dataset. Second, the authors are testing the AV1451 tracer, which was the same tracer used in the training set from the NFL-LONG dataset. So, this is not really a generalization task. Especially because the subject demographics are somewhat similar, this is less of a generalization result.

Similarly, the authors use a very small held out set of 10 subjects from NFL-LONG for testing. In my opinion, this is too small to make any meaningful conclusion against the baselines.

Further, on Figs. 2 and 3, the difference maps actually show that the synthesized images are fairly different from the true ones. Can the author comment on these differences?

I would also like the authors to run statistical tests to assess whether the differences in results are significant.

### Ablations
- Can the authors comment on why the MAE did not improve from some ablation studies?
- Were these ablations done on the 10 subjects in the NFL-LONG test set? If so, this is again too small of a sample size to make any conclusions
- Can the authors do one ablation with the standard GAN penalty and the gradient penalty?

**Questions:**

Many of my questions are embedded in the weaknesses section

What are some applications that you see this method being useful for?

---

### Meta-Review · Area_Chair_1wWz · 2025-12-25

**Summary:**

This paper received mixed reviews but mainly on negative side.  All reviewers have raised some concerns about this work.

**Reviewer Concerns:**

The main concerns relate to the clinical utility of the proposed approach, the lack of sufficient ablation studies, and limitations in the experimental setup.

**Reviewer Scores:**

Given the significance of the issues raised in the reviews, it is unlikely that the reviewers’ scores would have changed.

---

### Decision · Program_Chairs · 2026-01-26

Reject